# Evaluation of the Psychometric Properties of the Revised Piper Fatigue Scale in Patients with Multiple Sclerosis

**DOI:** 10.3390/healthcare10060985

**Published:** 2022-05-25

**Authors:** Anna Berardi, Giulia Graziosi, Gina Ferrazzano, Laura Casagrande Conti, Maria Grazia Grasso, Marco Tramontano, Antonella Conte, Giovanni Galeoto

**Affiliations:** 1Department of Human Neurosciences, Sapienza University of Rome, 001859 Rome, Italy; gina.ferrazzano@uniroma1.it (G.F.); antonella.conte@uniroma1.it (A.C.); 2School of Occupational Therapy, Sapienza University of Rome, 00185 Rome, Italy; giuliagraz99@gmail.com; 3Santa Lucia Foundation IRCCS, Via Ardeatina, 306, 00179 Rome, Italy; laura.casagrandeconti@gmail.com (L.C.C.); mg.grasso@hsantalucia.it (M.G.G.); m.tramontano@hsantalucia.it (M.T.); 4IRCCS Neuromed, 86077 Pozzilli, Italy

**Keywords:** assessment, fatigue, multiple sclerosis, rehabilitation, reliability, validity

## Abstract

Background: Fatigue is one of the most common symptoms in patients with multiple sclerosis (MS), and has a major impact on their quality of life. Measurement tools that assess the patient’s condition are commonly used in the neurological field, though diagnostic tools are currently unable to distinguish potential alternative causes of fatigue in individual patients. The Revised Piper Fatigue Scale (PFS-R) is a self-administered assessment scale that is internationally used for fatigue measurement. Aim: This study aimed to evaluate the reliability and validity of the Italian version of the PFS-R in patients with MS. Methods: Forty-one individuals were included in this study. Each participant in the study was given a dossier, in paper format, containing the informed consent form, a personal data sheet, the Fatigue Symptoms and Impacts Questionnaire in Relapsing Multiple Sclerosis (FSIQ-RMS), the Fatigue Severity Scale (FSS), and the PFS-R. Results: The PFS-R was found to have strong internal consistency, with a value of α equal to 0.977. Correlations between PFS-R, FSS, and FSIQ-RMS scores were analyzed using the Pearson correlation coefficient, and all scales showed statistically significant correlations. Conclusion: The PFS-R is a new self-administered tool to assess fatigue in patients with MS. It evaluates fatigue characteristics, difficulty in carrying out daily life activities, and how the individual feels as a result of this symptom. This tool was previously validated for use in cancer patients, which also allows us to make a comparison between different pathologies and rehabilitation treatments.

## 1. Introduction

Fatigue is one of the most common symptoms in patients with multiple sclerosis (MS) and significantly impacts patient quality of life [1]. Fatigue has been described in the literature as: “a sensation deriving from the difficulty in initiating or sustaining a voluntary effort” [2] or as “a sense of overwhelming and disproportionate fatigue (compared to the activity carried out)” [3], but also as a “feeling that refers to the lack of motivation to employ resources and engage in high-effort performance to cope with [the] situation” [4]. As with other neuropsychiatric symptoms, fatigue likely stems from several underlying causes. Physical fatigue can be a common reason for early retirement or sick leave, starting from the early stages of MS. Therefore, a timely and accurate diagnosis is essential [5]. Measurement tools to assess the patient’s condition are commonly used in the neurological field, though diagnostic tools are currently unable to distinguish the potential alternative causes of fatigue in individual patients [1]. To date, the most commonly used assessment tools are the Fatigue Severity Scale (FSS), the Unidimensional Fatigue Impact Scale (U-FIS), the Modified Fatigue Impact Scale (MFIS), the Neurological Fatigue Index (NFI-MS), the Fatigue Scale for Motor and Cognitive Functions (FSMC), the Multiple Sclerosis Quality of Life (MSQoL) Scale, and the Multiple Sclerosis International Quality of Life (MusiQoL) Scale [5,6].

The FSS was originally developed by Krupp et al. [7] to measure fatigue severity in a variety of medical conditions. The FSS is a simple and easy-to-use self-report questionnaire. It consists of nine statements that assess the severity of the patient’s fatigue symptoms in terms of how these symptoms affect motivation, exercise, physical function, and activities of daily living [8]. The MFIS is a modified version of the 40-item FIS. Individuals are required to rate how often fatigue has occurred during the previous week on a five-point scale (ranging from “never” to “all the time”). This scale was originally developed to evaluate the effects of fatigue on the quality of life in patients with chronic diseases, particularly MS [9].

All these evaluation scales are valid, reliable, and available in Italy, although a tool is not yet available to analyze the impact of fatigue on specific activities of daily life.

The Revised Piper Fatigue Scale (PFS-R) [10] is a self-administered assessment scale used internationally for fatigue measurement [11]. Over the years, it has been validated in several languages: English [11], French [12], Swedish [13], Portuguese [14], Thai [15], Dutch [16], Chinese (Hong Kong) [17], Italian [18,19], Korean [20], and Spanish [21]. The PFS-R was initially used for the assessment of fatigue in patients with cancer and was subsequently validated for other clinical settings, such as in chronic hepatitis B patients [20] and in patients undergoing maintenance dialysis [17]. In Italy, the PFS-R scale has been validated exclusively for the assessment of fatigue in patients with cancer. To date, no studies have been carried out on the validity of this scale in assessing fatigue in patients with MS, either in Italy or in other countries.

This scale provides information on the perception of fatigue and assesses the extent to which this symptom interferes in the individual’s usual daily life activities. The innovative aspect of this scale is that it investigates the difficulty of patients in carrying out certain daily life activities, such as social life, school or work activities, and sex life. Other items evaluate fatigue characteristics and how the patient feels as a result of fatigue. These aspects are necessary for the rehabilitation assessment of MS patients.

No studies have yet been carried out to evaluate the reliability and validity of this scale in assessing fatigue in people with MS. However, this is important as it would allow the evaluation of the same symptom in different pathologies; thus, the outcome of the same treatment (pharmacological or non-pharmacological) in patients with different pathologies could be compared. For example, previous studies have compared the physical activity in people with cancer and MS by evaluating the “psychosocial mechanism”; with the validation of PFS-R it would be possible to have another outcome measure to also compare the symptom of fatigue in different populations [22].

Therefore, this study aimed to evaluate the reliability and validity of the Italian version of the PFS-R in patients with MS.

## 2. Methods

This study was conducted by doctors and health professionals at Sapienza University of Rome and Policlinico Umberto I, in collaboration with the Santa Lucia Foundation in Rome [23,24,25,26,27,28,29,30,31,32].

### 2.1. Participants

Patients were recruited according to the following inclusion criteria: age over 18 and a diagnosis of MS [33]. Patients diagnosed with psychiatric pathologies were excluded from the study. Prior to recruitment, all participants were informed about the purpose and methods of the study and read and signed an informed consent form for the processing of personal data [34,35]. In the literature, sample size recommendations range from 2 to 20 subjects per item [36], and in the articles analyzed in a recent (2014) systematic review about sample size used to validate a scale, the mean subject to item ratio was 28, with a minimum of 1 and a maximum of 527 [37]. Furthermore, Shoukri et al. [38] report that, “However, in many cases, values of the reliability coefficient under the null and alternative hypotheses may be difficult to specify. Under such circumstances, one can safely recommend only two or three replications per subject”. For the present study, a minimum sample size of 39 was required according to previous validation of the tool and according to minimal recommendations from the Consensus-based Standards for the Selection of Health Measurement Instruments (COSMIN) [39]. The patients then completed the rating scales and a personal and clinical data collection form.

### 2.2. Procedures

The study was administered in the outpatient neurological clinic of Policlinico Umberto I in Rome and the Santa Lucia Foundation in Rome [40,41,42].

Each participant in the study was given a dossier, in paper format, containing an informed consent form, personal datasheet, and the FSIQ-RMS, FSS, and PFS-R. The Italian version of the tool used was already validated for individuals with cancer by Annunziata et al. [19].

### 2.3. Instruments

The original scale was published in 1989 by Piper et al. [10], with the subsequent revised version being reduced to 22 items [11]. The PFS-R describes four dimensions of subjective fatigue, according to the following subscales:(1)The behavioral severity subscale (6 items) evaluates the impact of fatigue on activities of daily life, such as work or school activities, social life, the sexual sphere, and hobbies. The first and sixth questions ask, respectively, how much discomfort the fatigue causes and how intense it is.(2)The cognitive severity subscale consists of elements related to mental state and mood. The last six items assess how the patient feels: patient/impatient; relaxed/tense; cheerful/depressed; able to concentrate/unable to concentrate; able to remember/unable to remember; and able to think clearly/unable to think clearly.(3)The affective meaning subscale (5 items) investigates how tired the patient is: pleasant/unpleasant; acceptable/unacceptable; protective/destructive; positive/negative; and normal/abnormal.(4)The sensory/emotional severity subscale (5 items) analyzes how the patient feels: strong/weak; awake/sleepy; full of life/apathetic; refreshed/tired; and full of energy/lacking in energy.

Each of the 22 items is assigned a score from 0 to 10, where 0 represents the best condition and 10 the worst. To obtain a subscale score, it is necessary to sum the scores of all the items in the subscale and divide this by the number of items. If there are missing data, but the interviewee answered at least 75–80% of the items, the average is calculated based on the number of items completed (e.g., 4 completed items: add their values and divide by 4). To calculate the PFS-R total score, the scores of the 22 items are added and divided by 22. Scores are interpreted as follows: 0 absent, 1–3 mild, 4–6 moderate, and 7–10 severe.

The FSS [7,43] is a simple and easy-to-use self-report questionnaire. It consists of nine statements that assess the severity of the patient’s fatigue symptoms in terms of how these symptoms affect motivation, exercise, physical function, and activities of daily living. Reflecting on their condition over the past two weeks, patients rate each item from 1 to 7 based on the extent to which they agree or disagree with each statement (1 = strongly disagree, 7 = strongly agree). The FSS can be assessed either by obtaining a total score or by calculating an average score on all nine items, with higher scores indicating more severe fatigue. The Italian version of the FSS showed optimal psychometric properties (Cronbach’s alpha = 0.95) [43].

The FSIQ-RMS [44] is a self-administered questionnaire consisting of 20 items divided into two domains. One domain evaluates symptoms related to fatigue and includes seven items, whereas the other domain is divided into three impact subscales: physical, cognitive/emotional, and coping (which is combined with two items of the physical impact subdomain). There is no single summary score in the FSIQ-RMS, only a symptom score and three impact subdomain scores. This tool showed good psychometric properties (Cronbach’s alpha values of 0.92) and it has recently been validated in Italian in a population of individuals with multiple sclerosis [45].

### 2.4. Statistical Analysis

Following the Consensus-Based Standards for the Selection of Health Status Measurement Instruments (COSMIN) checklist, internal consistency, construct validity, and cross-cultural validity of the scale were assessed [39]. The internal consistency of the PFS-R was examined by Cronbach’s alpha. Alpha values of 0.7, 0.8, and 0.9 are believed to represent a fair, good, and excellent degree of internal consistency, respectively [46]. Construct validity was measured using the Pearson correlation coefficient to determine the association between the PFS-R and the Italian versions of the FSS and FSIQ-RMS. Correlation coefficients of <0.30, <0.60, and ≥0.60 are considered to represent poor, moderate, and strong correlations, respectively. Cross-cultural validity was measured using the Pearson correlation coefficient to determine the association between PFS-R and sample demographics. Correlation coefficients of <0.30, <0.60, and ≥0.60 are considered to represent poor, moderate, and strong correlations, respectively. A p value less than or equal to 0.05 indicated statistical significance. The SPSS tool was used for statistical analysis.

## 3. Results

Individuals were recruited over a three-month period, from September to November 2021. The sample was composed of 41 patients. The average age was 48 years and 68.3% were female. The average body mass index (BMI) of recruited patients was 23.57 and all participants (100%) had relapsing-remitting MS, with an average time from diagnosis of 171.16 months. Secondary school was completed by 41.5% of the sample and 48.8% were employed. Furthermore, 48.8% of the recruited population attended a rehabilitation program. Patient demographic characteristics are shown in Table 1.

To verify the homogeneity between items, the internal consistency of the PFS-R was analyzed with Cronbach’s alpha (α). The evaluation scale had strong internal consistency, with a value of α equal to 0.977. The α value was also calculated in the various subscales. The first subscale (behavioral severity) had an α value equal to 0.96, the second subscale (cognitive severity) had an α value of 0.933, the third subscale (affective meaning) had an α value of 0.919, and the last subscale (sensory/emotional severity) had an α value of 0.942. As reported in Table 2, the value of Cronbach’s α for each subscale was always >0.7, demonstrating excellent internal consistency, i.e., an excellent interrelation between items. Furthermore, Table 2 demonstrates that all items were relevant, since if any one was eliminated the α value would decrease, consequently decreasing internal consistency.

During the first administration of the PFS-R, the participants also completed the FSS and FSIQ-RMS so that the construct validity of these scales could be calculated. The correlation between PFS-R, FSS, and FSIQ-RMS scores, reported in Table 3, was possible due to the calculation of the Pearson correlation coefficient. As reported in Table 3, the PFS-R significantly correlated with the FSS and FSIQ-RMS. Furthermore, as reported in Table 4, the correlation between PFS-R subscales and patient characteristics (BMI and months since diagnosis) was also statistically significant.

## 4. Discussion

This study aimed to evaluate the psychometric properties of the Italian version of the PFS-R in patients with MS to determine its validity and reliability.

Internal consistency was evaluated by calculating the value of Cronbach’s α, which was equal to 0.977. The α value was also calculated in the various subscales. The α value was equal to 0.96 in the first subscale (behavioral severity), 0.933 in the second subscale (cognitive severity), 0.919 in the third subscale (affective meaning), and 0.942 in the last subscale (sensory/emotional severity). These results are consistent with previous studies; in fact, the other studies in different cultures and different diseases all reported an alpha value > than 80 [13,16,19,20,21].

When administered to patients with MS, the Italian version of the PFS-R had excellent internal consistency, with an α value well above 0.7. Furthermore, these data were consistent with the validation of PFS-R in cancer, on which this study was based and in which an internal consistency value of 0.952 was observed [19]. To analyze the convergent validity of the PFS-R, all participants also completed the FSS and FSIQ-RMS. The correlation between patient scores on these scales was measured by the Pearson correlation coefficient, as reported in Table 3. PFS-R scores significantly correlated with FSS and FSIQ-RMS scores. This is the first study analyzing the correlation of this scale with FSS and FSIQ-RMS; however, previous studies found significant correlations with the Cancer Fatigue Scale, Beck Depression Inventory, Verbal Numerical Rating Scale, and Karnofsky Performance Scale. [47] The Italian version validated for cancer patients showed correlations with the European Organisation for Research and Treatment of Cancer (EORTC) QLQ-C30 [19].

This study had certain limits; in fact, the sample size did not allow for the examination of the influence of the various clinical and sociodemographic variables on fatigue, such as different EDSS or MS types.

In conclusion, the PFS-R is a new self-administered tool to assess fatigue in patients with MS. It evaluates fatigue characteristics, difficulty in performing daily life activities, and how the individual feels as a result of fatigue. This tool was previously validated in cancer, which allowed us to make a comparison between different pathologies and rehabilitation treatments.

## Figures and Tables

**Table 1 healthcare-10-00985-t001:** Patient demographic characteristics.

		Mean	Standard Deviation
	Age	48.00	9.808
	Body mass index	23.57	4.55
	Months from diagnosis	171.16	224.7
		Frequency	Percentage
Sex	Male	13	31.7
	Female	28	68.3
MS type	Relapsing-remitting	41	100
Education	Elementary	8	19.5
	Diploma	17	41.5
	Bachelor’s degree	4	9.8
	Master’s degree	12	29.3
Employment	No	21	51.2
	Yes	20	48.8
Rehabilitation	No	21	51.2
	Yes	20	48.8

**Table 2 healthcare-10-00985-t002:** Internal consistency: Cronbach’s alpha for the subscales and with items deleted.

	Mean	Standard Deviation	Cronbach’s Alpha if Item Deleted
Behavioral severity			
Item 1	3.86	3.102	0.953
Item 2	4.35	2.993	0.948
Item 3	4.19	3.373	0.947
Item 4	4.68	3.667	0.974
Item 5	4.57	3.254	0.946
Item 6	4.38	2.994	0.947
Cronbach’s alpha first subscale	0.96		
Cognitive severity			
Item 13	4.41	2.855	0.946
Item 20	4.10	2.888	0.889
Item 21	4.00	3.138	0.921
Item 22	3.41	2.966	0.894
Cronbach’s alpha second subscale	0.933		
Affective meaning			
Item 10	5.71	3.118	0.896
Item 9	5.34	3.034	0.892
Item 11	5.61	3.167	0.903
Item 8	4.97	3.062	0.895
Item 17	4.00	2.986	0.937
Item 18	4.76	2.954	0.926
Item 7	5.18	3.344	0.895
Cronbach’s alpha third subscale	0.919		
Sensory/emotional severity			
Item 14	4.58	2.960	0.920
Item 15	5.13	2.643	0.924
Item 16	5.40	2.509	0.921
Item 12	5.13	2.462	0.924
Item 19	4.55	2.773	0.949
Cronbach’s alpha fourth subscale	0.942		
Cronbach’s alpha total scale	0.977		

**Table 3 healthcare-10-00985-t003:** Construct validity: Pearson correlation coefficient between the Fatigue Symptoms and Impacts Questionnaire in Relapsing Multiple Sclerosis (FSIQ-RMS), the Fatigue Severity Scale (FSS), and the Revised Piper Fatigue Scale (PFS-R).

	FSIQ-RMS Total 24 h	FSIQ-RMS Total 7 d	FSIQ-RMS Fatigue	FSIQ-RMS Physical Impact	FSIQ-RMS Cognitive and Emotional Impact	FSIQ-RMS Coping Impact	Total FSS
PFS-R behavioral severity	0.859 **	0.821 **	0.859 **	0.770 **	0.755 **	0.797 **	0.628 **
PFS-R cognitive severity	0.719 **	0.799 **	0.719 **	0.709 **	0.877 **	0.670 **	0.490 **
PFS-R affective meaning	0.824 **	0.792 **	0.824 **	0.689 **	0.772 **	0.767 **	0.546 **
PFS-R sensory/emotional severity	0.769 **	0.814 **	0.769 **	0.713 **	0.812 **	0.767 **	0.423 *
Total PFS-R	0.859 **	0.872 **	0.859 **	0.781 **	0.868 **	0.812 **	0.568 **

** The correlation was significant at the level of 0.01 (two-tailed). * The correlation was significant at the level of 0.05 (two-tailed).

**Table 4 healthcare-10-00985-t004:** Cross-cultural validity: Pearson correlation coefficient between the Revised Piper Fatigue Scale (PFS-R) and patient demographic characteristics.

	Body Mass Index	Months from Diagnosis
PFS-R behavioral severity	0.409 **	0.151
PFS-R cognitive severity	0.308	0.199
PFS-R affective meaning	0.268	0.28
PFS-R sensory/emotional severity	0.345 *	0.164
Total PFS-R	0.362 *	0.212

** The correlation was significant at the level of 0.01 (two-tailed). * The correlation was significant at the level of 0.05 (two-tailed).

## Data Availability

The data that support the findings of this study are available from the corresponding author upon reasonable request.

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
