# Peer review of "Evaluation of the Psychometric Properties of the Revised Piper Fatigue Scale in Patients with Multiple Sclerosis"

_healthcare, 2022, doi:10.3390/healthcare10060985_

Round 1

Reviewer 1 Report

Here, the authors describe the evaluation and validation of an Italian translation of the Revised Piper Fatigue Scale (PFS-R) in a sample of 41 MS patients in Italy. The PFS-R had strong internal consistency over 3 months’ follow-up and was significantly correlated with other fatigue measures assessed, including the FSS and FSIQ-RMS.

  • Please add sample size to the Abstract.
  • Please specify MS type breakdown in text and in Table 1. I presume it’s all just RRMS but please specify.
  • Please use sex and male/female terminology instead of gender and men/women.

Author Response

Reviewer Comment

Response

Line #

Reviewer #1

Please add sample size to the Abstract.

 Sample size has been added to abstract

18

Please specify MS type breakdown in text and in Table 1. I presume it’s all just RRMS but please specify

MS type has been added

137; table 1

Please use sex and male/female terminology instead of gender and men/women.

Terminology have been corrected

Table 1

Reviewer 2 Report

The authors have made an important effort for evaluate the psychometric properties of the Revised Piper Fatigue Scale in patients with Multiple Sclerosis. Although the objective of the article is interesting, but it is necessary to improve some sections of the article. Specifically of the methodology.

For example, I would like to know the reason why you have not calculated the sample size necessary to carry out your study. In fact, a sample size of 41 patients for conducting this study was too low.

I recognize the importance of adapting it to a specific patient sample. Even so, I consider that they should have used a control sample and, in this way, compare the scores.

They have put a reference as a diagnostic criterion for sclerosis. Even so, I would like them to provide more information on the characteristics of the patients employed. Information provided in table 1 was scarce.

It strikes me that the code assigned by the Ethics Committee of the reference University/Hospital/Research entity to carry out this study does not appear. In fact, they did not provide this con

It would be important to know if the authors have had to make any modifications to the questionnaire validated in Italian with respect to the previously validated version.

It would be important to know the moment in which the study was carried out, since the COVID-19 pandemic has created an exceptional situation that could affect the results.

It is essential to add the psychometric properties of the rest of the questionnaires used in this study (e.g., FSS and FSIQ-RMS).

Absence of control for multiple comparisons (e.g., Bonferroni correction).

Finally, the introduction seems to me insufficient to justify the importance of validating this scale for MS. Specifically, they start from a symptom that is common to many diseases and, therefore, does not sufficiently justify using this scale.

Author Response

Reviewer Comment

Response

Line #

Reviewer #2

For example, I would like to know the reason why you have not calculated the sample size necessary to carry out your study. In fact, a sample size of 41 patients for conducting this study was too low.

Information about sample size have been added

82-85

I recognize the importance of adapting it to a specific patient sample. Even so, I consider that they should have used a control sample and, in this way, compare the scores.

Usually in in the validation of outcome measure is not used a comparison sample, this is the step after validating the toll. However we agree with reviewer in comparing results of our study with previous ones in other diseases. This information has been added to discussion

165-167; 174-178

They have put a reference as a diagnostic criterion for sclerosis. Even so, I would like them to provide more information on the characteristics of the patients employed. Information provided in table 1 was scarce.

We added MS type, but unfortunately we are not able to get other information

137; table 1

It strikes me that the code assigned by the Ethics Committee of the reference University/Hospital/Research entity to carry out this study does not appear. In fact, they did not provide this con

“Ethics committee approval is not required for this study, this research involve secondary use of clinical data which is provided without any identifier or group of identifiers which would allow attribution of private information to an individual. Informed consent was obtained from all participants for being included in the study.”

This has been reported in the Ethic statement

188-191

It would be important to know if the authors have had to make any modifications to the questionnaire validated in Italian with respect to the previously validated version.

This was not a study of translation and cultural adaptation. The Italian version of the tool is already validated for the cancer population. We used that version

90, 91

It would be important to know the moment in which the study was carried out, since the COVID-19 pandemic has created an exceptional situation that could affect the results.

The study was conducted from September to November 2021 as reported in lines 140, 141

It is essential to add the psychometric properties of the rest of the questionnaires used in this study (e.g., FSS and FSIQ-RMS).

Psychometric properties of the tools have been added

119, 120; 125,126

Absence of control for multiple comparisons (e.g., Bonferroni correction)..

We believe that this kind of analysis is not appropriate for this study design and according with guidelines for validation tools (COSMIN)

Finally, the introduction seems to me insufficient to justify the importance of validating this scale for MS. Specifically, they start from a symptom that is common to many diseases and, therefore, does not sufficiently justify using this scale

Rationale has been improved

72-74

Reviewer 3 Report

I reviewed the manuscript entitled " Evaluation of the psychometric properties of the Revised Piper Fatigue Scale in patients with Multiple Sclerosis". In this paper, a validated questionnaire for fatigue assessment has been validated in the Italian language in MS patients. There are some concerns:

Minor concern: The abstract should be separated like the entire body of the article into different sections (introduction, methods,...)

Major concern: The discussion is too short. There are sufficient articles that should be reviewed in this part. 

Author Response

Reviewer Comment

Response

Line #

Reviewer #3

The abstract should be separated like the entire body of the article into different sections (introduction, methods,...)

Headings have been added to the abstract

11; 16; 18; 22; 25

The discussion is too short. There are sufficient articles that should be reviewed in this part.

Discussion have been enlarged

Throughout the discussion

Round 2

Reviewer 2 Report

The authors have tried to answer the criticism received about the previous version. Even so, the authors have provided a series of statements that are extremely brief and in which they have gone into minimal depth. Therefore, I would like to know if they could be given more time to respond to the criticisms, since in the current state I still recommend that the article be rejected, since they have not properly justified the comments. I will explain my comment below:

1) According to the authors, they have justified the need for a sample size of 39 participants by adding a bibliographical reference. To calculate the sample size, it is necessary to specify the population on which it is based (e.g., population size), confidence level, margin of error, population proportion, among others.

2) To justify the psychometric properties of the version they have used, they have simply specified that it has good psychometric properties (e.g., This tool showed good psychometric properties and it is being validated in Italian). What I request is the calculation of the Cronbach’s alpha for the version you have used. In addition, if there are previous studies that justify its validity for this type of population or similar.

3) They consider that adding a sentence after the objective of the study justifies the importance of carrying out this study. In fact, this does not justify the use of a questionnaire to assess a symptom that is common to any type of disorder and, furthermore, is not an important or apparently important symptom for this disorder. Therefore, I would request that you review in DEPTH to justify its use.

4) They did not offer enough information regarding recruitment process as well as demographic characteristics, potential confounding of the variables included.

5) They did not apply statistical control for multiple comparisons.

6) Although this is a part of a previous study, it is completely necessary to specify Ethics Committee assigned code.   

Author Response

Date: May 09th, 2022

Dear Editor,

We appreciate the opportunity to resubmit our article entitled “Evaluation of the psychometric properties of the Revised Piper Fatigue Scale in patients with Multiple Sclerosis.” We would like to thank the referees for the careful and constructive reviews. We have made corresponding changes directly to the manuscript where appropriate with changes tracked. The revised version of our manuscript accompanies this letter. All comments by the reviewer have been addressed. Based on his/her comments, we have made changes to the manuscript, which are detailed below.

Reviewer Comment

Response

Line #

Reviewer #2

1) According to the authors, they have justified the need for a sample size of 39 participants by adding a bibliographical reference. To calculate the sample size, it is necessary to specify the population on which it is based (e.g., population size), confidence level, margin of error, population proportion, among others.

This calculation is necessary for experimental studies. For validation studies the sample size depends on the number of items of the tool.

In literature, sample size recommendations range from 2 to 20 subjects per item (21), in the articles analyzed in a recent (2014) systematic review about sample size used to validate a scale, the mean subject to item ratio was 28, with a minimum of 1 and a maximum of 527.(22) Furthermore, Shoukri et al. (23) reports that “However, in many cases, values of the reliability coefficient under the null and alternative hypotheses may be difficult to specify. Under such circumstances, one can safely recommend only two or three replications per subject”.

21. Hair JE, Anderson RE, Tatham RL, Black WC. Multivariate Data Analysis: With Readings. Englewood Cliffs, NJ: Prentice -Hall; 1995. p. 757.

22. Anthoine E, Moret L, Regnault A, Sébille V, Hardouin JB. Sample size used to validate a scale: a review of publications on newly-developed patient reported outcomes measures. Health Qual Life Outcomes. 2014; 12: 2. doi:  [10.1186/s12955-014-0176-2]

23. Shoukri MM, Asyali MH, Donner A. Sample size requirements for the design of reliability

study: review and new results. Stat Methods Med Res 2004; 13: 251-71.

86-92

2) To justify the psychometric properties of the version they have used, they have simply specified that it has good psychometric properties (e.g., This tool showed good psychometric properties and it is being validated in Italian). What I request is the calculation of the Cronbach’s alpha for the version you have used. In addition, if there are previous studies that justify its validity for this type of population or similar.

This tool showed good psychometric properties (Cronbach’s alpha values of 0.92) and it has recently been validate in Italian in a population of individuals with Multiple Sclerosis. Ruotolo I, Sellitto G, Ianniello A, Petsas N, Castelli L, Galeoto G, Berardi A, Barletta V, Conte A, Pozzilli C. Italian translation and validation of fatigue symptoms and impacts questionnaire in relapsing multiple sclerosis (FSIQ-RMS). Neurol Sci. 2022 Apr 22. doi: 10.1007/s10072-022-06080-1. Epub ahead of print. PMID: 35451663.

134-136

3) They consider that adding a sentence after the objective of the study justifies the importance of carrying out this study. In fact, this does not justify the use of a questionnaire to assess a symptom that is common to any type of disorder and, furthermore, is not an important or apparently important symptom for this disorder. Therefore, I would request that you review in DEPTH to justify its use.

The reason why using this outcome measure also in people with MS and not only in people with cancer is that there are other studies analyzing the pshycosocial mechanism in these two population, thanks to our study we now have an additional tool to measure the fatigue in these populations (and others) and compare results.

McAuley E, White SM, Rogers LQ, Motl RW, Courneya KS. Physical activity and fatigue in breast cancer and multiple sclerosis: psychosocial mechanisms. Psychosom Med. 2010 Jan;72(1):88-96. doi: 10.1097/PSY.0b013e3181c68157. Epub 2009 Nov 30. PMID: 19949160; PMCID: PMC2825155.

74-77

4) They did not offer enough information regarding recruitment process as well as demographic characteristics, potential confounding of the variables included.

This study has certain limits, in fact the sample size did not allow to evaluate different expression of fatigue in subpopulation, such as different EDSS or MS type.

192, 193

5) They did not apply statistical control for multiple comparisons.

We believe that this kind of analysis is not appropriate for this study design and according with guidelines for validation tools (COSMIN). The Bonferroni Correction is required in the outcome research and RCT. The reduction of significance through the Bonferroni Correction allows to compare different outcome measures when it is evaluated the responsiveness. The responsiveness was not the primary objective of our study.

In our study is not appropriate to have a control group (for example healthy subjects) since the fatigue symptom si directly correlated to a clinical condition, it does not refer to physical tiredness.

6) Although this is a part of a previous study, it is completely necessary to specify Ethics Committee assigned code.  

Ethics Committee assigned code.    Has been added

204-206

Reviewer #3

All of the remarks have been correctly revised.

We hope that the new version of our manuscript is acceptable for publication.

Best regards,

Anna Berardi

Reviewer 3 Report

All of the remarks have been correctly revised. 

Author Response

Date: May 09th, 2022

Dear Editor,

We appreciate the opportunity to resubmit our article entitled “Evaluation of the psychometric properties of the Revised Piper Fatigue Scale in patients with Multiple Sclerosis.” We would like to thank the referees for the careful and constructive reviews. We have made corresponding changes directly to the manuscript where appropriate with changes tracked. The revised version of our manuscript accompanies this letter. All comments by the reviewer have been addressed. Based on his/her comments, we have made changes to the manuscript, which are detailed below.

Reviewer Comment

Response

Line #

Reviewer #2

1) According to the authors, they have justified the need for a sample size of 39 participants by adding a bibliographical reference. To calculate the sample size, it is necessary to specify the population on which it is based (e.g., population size), confidence level, margin of error, population proportion, among others.

This calculation is necessary for experimental studies. For validation studies the sample size depends on the number of items of the tool.

In literature, sample size recommendations range from 2 to 20 subjects per item (21), in the articles analyzed in a recent (2014) systematic review about sample size used to validate a scale, the mean subject to item ratio was 28, with a minimum of 1 and a maximum of 527.(22) Furthermore, Shoukri et al. (23) reports that “However, in many cases, values of the reliability coefficient under the null and alternative hypotheses may be difficult to specify. Under such circumstances, one can safely recommend only two or three replications per subject”.

21. Hair JE, Anderson RE, Tatham RL, Black WC. Multivariate Data Analysis: With Readings. Englewood Cliffs, NJ: Prentice -Hall; 1995. p. 757.

22. Anthoine E, Moret L, Regnault A, Sébille V, Hardouin JB. Sample size used to validate a scale: a review of publications on newly-developed patient reported outcomes measures. Health Qual Life Outcomes. 2014; 12: 2. doi:  [10.1186/s12955-014-0176-2]

23. Shoukri MM, Asyali MH, Donner A. Sample size requirements for the design of reliability

study: review and new results. Stat Methods Med Res 2004; 13: 251-71.

86-92

2) To justify the psychometric properties of the version they have used, they have simply specified that it has good psychometric properties (e.g., This tool showed good psychometric properties and it is being validated in Italian). What I request is the calculation of the Cronbach’s alpha for the version you have used. In addition, if there are previous studies that justify its validity for this type of population or similar.

This tool showed good psychometric properties (Cronbach’s alpha values of 0.92) and it has recently been validate in Italian in a population of individuals with Multiple Sclerosis. Ruotolo I, Sellitto G, Ianniello A, Petsas N, Castelli L, Galeoto G, Berardi A, Barletta V, Conte A, Pozzilli C. Italian translation and validation of fatigue symptoms and impacts questionnaire in relapsing multiple sclerosis (FSIQ-RMS). Neurol Sci. 2022 Apr 22. doi: 10.1007/s10072-022-06080-1. Epub ahead of print. PMID: 35451663.

134-136

3) They consider that adding a sentence after the objective of the study justifies the importance of carrying out this study. In fact, this does not justify the use of a questionnaire to assess a symptom that is common to any type of disorder and, furthermore, is not an important or apparently important symptom for this disorder. Therefore, I would request that you review in DEPTH to justify its use.

The reason why using this outcome measure also in people with MS and not only in people with cancer is that there are other studies analyzing the pshycosocial mechanism in these two population, thanks to our study we now have an additional tool to measure the fatigue in these populations (and others) and compare results.

McAuley E, White SM, Rogers LQ, Motl RW, Courneya KS. Physical activity and fatigue in breast cancer and multiple sclerosis: psychosocial mechanisms. Psychosom Med. 2010 Jan;72(1):88-96. doi: 10.1097/PSY.0b013e3181c68157. Epub 2009 Nov 30. PMID: 19949160; PMCID: PMC2825155.

74-77

4) They did not offer enough information regarding recruitment process as well as demographic characteristics, potential confounding of the variables included.

This study has certain limits, in fact the sample size did not allow to evaluate different expression of fatigue in subpopulation, such as different EDSS or MS type.

192, 193

5) They did not apply statistical control for multiple comparisons.

We believe that this kind of analysis is not appropriate for this study design and according with guidelines for validation tools (COSMIN). The Bonferroni Correction is required in the outcome research and RCT. The reduction of significance through the Bonferroni Correction allows to compare different outcome measures when it is evaluated the responsiveness. The responsiveness was not the primary objective of our study.

In our study is not appropriate to have a control group (for example healthy subjects) since the fatigue symptom si directly correlated to a clinical condition, it does not refer to physical tiredness.

6) Although this is a part of a previous study, it is completely necessary to specify Ethics Committee assigned code.  

Ethics Committee assigned code.    Has been added

204-206

Reviewer #3

All of the remarks have been correctly revised.

We hope that the new version of our manuscript is acceptable for publication.

Best regards,

Anna BErardi
